# Individual differences and the transposed letter effect during reading

**Charlotte E. Lee**[1]*, **Ascensión Pagán**[2], **Hayward J. Godwin**[1], **Denis Drieghe**[1]

1 School of Psychology, University of Southampton, Southampton, United Kingdom, 2 School of Psychology and Vision Sciences, University of Leicester, Leicester, United Kingdom

* c.lee@soton.ac.uk

**Data Availability Statement:** All data and materials are available online at: https://osf.io/b2rdm/.

**Funding:** CL was funded by the UKRI Economic and Social Research Council South Coast Doctoral

## Abstract

When a preview contains substituted letters (SL; markey) word identification is more disrupted for a target word (monkey), compared to when the preview contains transposed letters (TL; mnokey). The transposed letter effect demonstrates that letter positions are encoded more flexibly than letter identities, and is a robust finding in adults. However, letter position encoding has been shown to gradually become more flexible as reading skills develop. It is unclear whether letter position encoding flexibility reaches maturation in skilled adult readers, or whether some differences in the magnitude of the TL effect remain in relation to individual differences in cognitive skills. We examined 100 skilled adult readers who read sentences containing a correct, TL or SL preview. Previews were replaced by the correct target word when the reader's gaze triggered an invisible boundary. Cognitive skills were assessed and grouped based on overlapping variance via Principal Components Analysis (PCA) and subsequently used to predict eye movement measures for each condition. Consistent with previous literature, adult readers were found to generally encode letter position more flexibly than letter identity. Very few differences were found in the magnitude of TL effects between adults based on individual differences in cognitive skills. The flexibility of letter position encoding appears to reach maturation (or near maturation) in skilled adult readers.

## Introduction

A large body of evidence suggests that eye movements during reading are fundamentally linked to a reader's cognitive processing and reveal processing difficulties related to features of the text (see [1,2]). A number of studies have also looked at the influence of individual differences in reading skills, and how these relate to the patterns of eye movement behaviour [3–14]. Evidence suggests that skilled adult readers process words more quickly than less skilled readers or children, as seen in shorter fixations, shorter gaze durations and fewer refixations (see [1] for a review).

The average skilled adult reader can extract information such as spacing from 14–15 character spaces from the point of fixation in the direction of the upcoming word (rightwards in English readers) and 3–4 character spaces in the direction of the previous word in alphabetic

Training Partnership (Grant Number ES/P000673/
1). This work formed part of a PhD with the South
Coast Doctoral Training Partnership (https://
southcoastdtp.ac.uk/). The funders had no role in
study design, data collection and analysis, decision
to publish, or preparation of the manuscript.

**Competing interests:** The authors have declared
that no competing interests exist.

languages [15]. This asymmetric visual field is a phenomenon which facilitates the pre-processing of information related to upcoming words. Word identification, which requires higher visual acuity, occurs 3–4 letters to the left and 6–7 letters to the right of fixation in alphabetic languages that are read from left to right [16,17]. In practice, the size of the perceptual span varies between readers, and notably increases with age until skilled reading is fully developed [18–20]. Though differences in the size of the perceptual span can relate to differences in text processing difficulty (as the difficulty of a text increases, the readers' perceptual span shrinks [18]), individual differences are also suggested to influence the size of a reader's perceptual span. Veldre and Andrews [12] found that adult readers with high spelling and reading abilities had larger perceptual spans during reading than readers with low spelling and reading abilities. A study by Häikiö et al. [8] found that slower readers identified fewer letters during a fixation than faster readers of the same age group (for Finnish children aged 8, 10 and 12, and for adults). They suggested that slower readers, unlike faster readers, allocate most of their processing resources to words when they are directly fixated on the fovea (2 degrees in the centre of vision). However, readers generally also process some information about an upcoming word parafoveally (in the parafoveal region, approximately 5 degrees to the left or right of fixation) when the eyes are fixating the preceding word.

## Parafoveal processing in skilled reading

Parafoveal processing enables the reader to extract information about the upcoming word before it is directly fixated, and when this information is useful, the upcoming word is processed more rapidly. Studies that explore parafoveal processing during reading most often use a gaze contingent invisible boundary paradigm [15,21]. This paradigm allows researchers to display a manipulated version of the target word to the right of a reader's gaze until their eyes cross an invisible boundary, whereupon the display is switched to show the correct target. The change occurs during a saccade when the readers' vision is blurred, resulting in this manipulation being usually undetected. Studies have found that reading is facilitated when the preview of the target word is identical to the target word [21]. In addition, when a preview shares orthographic or phonological information with the target word some preview benefit is also found, where faster processing of the parafoveal word when it is subsequently fixated is facilitated by information gathered from the preview word [22].

## Individual differences in parafoveal processing

Individual differences in reading and spelling abilities have been found to modulate the amount of information that can be extracted from the parafovea and used to facilitate word identification [13,14,23,24]. Skilled readers who are also good spellers extract more parafoveal information about word length and lexical features of a word [13,14]. However, differential effects have been noted for the extraction of semantic information. Good spellers had a reduced preview benefit from semantically related previews (demonstrating competition between semantic and orthographic information), whereas high reading ability has been found to predict a greater benefit from semantically related previews [23].

The current study further investigated individual differences in parafoveal processing of orthographic information in an upcoming word. Our focus was on individual differences in the extraction of letter position and letter identity information in the parafovea. To explore this, we first need to discuss how this information is encoded in isolated word identification.

## The transposed letter effect

To 'crack the orthographic code' research has used transposed letter (TL) stimuli, where the position of letters in a real word are swapped to create a nonword, to investigate how precisely letter position information is encoded. Priming paradigms and lexical decision tasks are used to see how word identification is affected by TL pseudowords compared to pseudowords created by substituting letters (SL) within a base word. Importantly, researchers are able to answer distinct research questions with each of these paradigms [25].

In lexical decision tasks, participants must decide whether a letter string is a real word. Using this paradigm, researchers can investigate to what extent a TL or SL pseudoword activates lexical information. As a result, it has been shown that a TL pseudoword increases the response latency and likelihood of errors (misinterpreting the pseudoword as a real word) compared to a SL pseudoword [26–30]. In other words, a TL pseudoword is more difficult to reject compared to an SL pseudoword because it is perceived to be more similar to the real base word.

Alternatively, in masked priming studies, the relationship between a prime and a target word is manipulated. Using this technique, researchers can examine the amount of disruption for processing the target word when an orthographic preview of a TL or SL prime is already activated. Findings from such research has demonstrated that the time to identify a real word target (judge) is reduced when a TL prime (jugde) is used compared to a substituted letter (SL) prime (junpe) [e.g., 31–34]. Again, TL pseudowords are perceived to be more similar to the real base word (target word) than SL pseudowords, and as a result processing of the target word is less disrupted following a TL prime compared to an SL prime.

Both paradigms have provided evidence that is consistent with the idea that a TL pseudoword is perceived to be more similar to the base word than a SL pseudoword is. These findings suggest that letter positions and letter identities are encoded independently, given that there is a processing advantage when letter identities are preserved despite changes in letter positions.

This flexible letter position encoding mechanism has been incorporated in recent models of word recognition (the SOLAR model [35,36], the Open Bigram model [28,37–39], the Overlap model [40], the SERIOL model [41] and the Bayesian Reader [42]).

Sentence-reading studies have also demonstrated effects consistent with the evidence using isolated word recognition paradigms. During silent sentence reading, when a target word has been replaced by a misspelled word (with no display change) in foveal vision, transposed-letter effects are inferred from the relative ease of processing a TL pseudoword, compared to an SL pseudoword. Rayner et al. [43] observed that readers' eye movements were only slightly disrupted when reading a sentence containing a TL pseudoword.

In a parafoveal preview experiment, researchers can determine the extent to which information gained from a misspelled preview can be rapidly integrated with a correct target word following a display change. Since the preview word is only accessible in the parafovea, this process takes place during a very early stage of word processing. Transposed letter effects have been investigated using the boundary paradigm and have consistently found preview benefits for TL pseudoword previews compared to SL previews [44–47]. Given the evidence from these studies, there is consensus that letter position information is encoded flexibly in skilled readers [36,41,48], however, since the extraction and use of parafoveal information is influenced by individual differences in skilled adult readers [13,14,23,24], there may be similar modulation of the transposed letter effect in parafoveal preview.

## Individual differences in children's letter position encoding

The aim of the current study was to assess how individual differences in cognitive skills may influence how letter position information is processed in skilled adult readers. Relatively little

research in this field has focussed on individual differences within adult readers, though changes in the magnitude of transposed letter effects have been observed in relation to children's reading abilities [49–53]. Pagán et al. [49] investigated the position of a transposition within a word in a reading-like task and noted that the amount of disruption for a misspelled word with a transposition of the 2nd and 3rd characters was smaller for children with higher reading skills than for children with lower reading skills. Using a lexical decision task, Gómez et al. [50] also found that in 6th grade Catalan children, individual differences in reading ability, specifically in pseudoword reading (measured by a subtest from PROLEC-R [54]) modulated transposed letter effects. Better readers were less likely to confuse TL pseudowords (mohter) with the real base word (mother) than less skilled readers. Negligible differences were associated with word-reading (measured by a subtest from PROLEC-R [54]) and perceptual processing speed (measured by a symbol search subset of the Wechsler Intelligence Scale for Children [55]).

Similarly, Hasenäcker and Schroeder [51] found that children's orthographic knowledge (a composite score calculated using a principal components analysis (PCA) of scores on spelling [56], vocabulary [57] and a word-reading to nonword-reading difference score [58] modulated transposed letter effects within grades in a longitudinal study of German children from grade 2 to 4. The cost associated with an SL prime was larger for children with higher levels of orthographic knowledge than for those with lower levels of orthographic knowledge, whereas there was no significant cost for a TL prime at any level of orthographic knowledge. Importantly, Hasenäcker and Schroeder [51] noted that the modulation associated with orthographic knowledge was similar to the modulation observed across grades. They concluded that whilst letter position encoding for words is fairly flexible in early readers, it becomes more flexible as reading skills are developed. They suggested that such changes are driven by increasing orthographic knowledge in children, for which grade is a good proxy. These investigations demonstrate that letter position encoding becomes more flexible as reading skills improve [49–51]. However, it remains unclear whether differences within skilled adult readers remain in relation to individual differences or whether letter position encoding is stable in this population. Next, we consider a model of visual word recognition that discusses letter position encoding in relation to children's reading development, before discussing how it may relate to individual differences in skilled adult readers.

## Orthographic processing during reading development

The Multiple-route model [39] suggests that the precise positions of letters are important whilst children decode written words phonologically, translating letters to sounds. As children develop reading skills they rely less on this process, and begin to use orthographic processes, bypassing the need to directly convert letters to sounds. The model includes two orthographic routes to achieve this: a fine-grained route where the coding of letter sequences is location-specific, and a coarse-grained route that uses non-continuous-location-invariant bigrams. For example, the word FARM can be coded by the bigrams FA, FR, FM, AR, AM, RM. Therefore, according to this model, an increased reliance on the coarse-grained route as reading skills develop leads to more flexibility in letter position encoding.

Though this model focuses on children's reading development rather than individual differences in adults, some predictions can be adapted for the current study. If skilled readers continue to rely on a coarse-grained route to orthographic decoding, the impact of a transposed letter preview may be stable across skilled adult readers, reflecting a maturation of letter position encoding flexibility. However, given that many cognitive skills remain variable in skilled adults [59–61], individual differences in cognitive skills may continue to predict differences in

the flexibility of letter position encoding once skilled reading is achieved. Similar to differences in the magnitude of the transposed letter effect seen in developing children related to reading ability [49,51], the effect of a transposed letter may be modulated by individual differences in adults. Extraction and use of this information during parafoveal processing may be greater for adults with better reading and spelling abilities as observed by Veldre and Andrews [13,14].

## Individual differences in skilled adult readers

A few studies have investigated a range of individual differences within adult readers' orthographic processing, though, to our knowledge, none have specifically investigated transposed letter effects in this way during reading. Andrews and Lo [62] used masked priming to investigate individual differences in reading ability, spelling [63] and vocabulary [63] in adult readers. They found that high reading, spelling, and vocabulary skills (calculated as a composite score based on shared variance) was associated with stronger facilitation from an orthographic nonword prime (different from the target in any single letter position). Welcome and Trammel [64] found comparable patterns associated with phonemic decoding efficiency scores [65] where adults with lower scores showed a general benefit of orthographic relatedness (both pronounceable and unpronounceable anagram primes were facilitatory for both word and non-word targets). This study measured verbal IQ [66], sight word efficiency [65], phonemic decoding efficiency (nonword reading) [65], print exposure [67] an orthographic choice task [68], a wordlikeness task [69] and an adult reading history questionnaire [70]. Adults with higher phonemic decoding efficiency scores benefitted only in conditions where pronounceable primes were used for word targets. Though these studies did not investigate effects of TL nonword primes specifically, they suggest that differences in orthographic priming may occur in relation to individual differences in these skills. In an investigation of individual differences in masked form priming Adelman et al. [71] found that those with strong spelling abilities and large vocabularies had faster response times and were less susceptible to priming in general than less skilled spellers and those with smaller vocabularies. This is similar to patterns seen in relation to reading and spelling abilities in parafoveal preview benefit [13–14,23–24].

The current study utilised a sentence reading task with an invisible boundary paradigm to explore transposed letter effects in comparison to a large battery of cognitive tasks. Consistent with previous evidence using masked priming and parafoveal preview paradigms, if the letter position encoding mechanism varies in adult readers, we predicted that spelling and word naming scores would modulate the size of the transposed letter effect during parafoveal processing. There may also be other cognitive skills that play a role, for example, Kuperman and Van Dyke [4] found that individual differences in Rapid Automatized Naming (RAN) and word identification were the two most reliable measures when predicting eye movements during reading when assessing a large test battery.

The aim of the current study was to investigate whether letter position encoding during parafoveal processing matures in skilled adult readers, or whether individual differences in a range of cognitive skills influence the parafoveal processing of a TL nonword preview in comparison to a SL nonword preview. The following tests were included. First, two commonly used reading ability tests that differ in subtest components were selected; the Wechsler Individual Achievement Test (WIAT-II [72]), which features reading comprehension, word reading and pseudoword decoding; and the Nelson Denny Reading Test (NDRT [73]), which includes a measure of vocabulary and reading comprehension. Since word reading and pseudoword decoding are measures seen in previous investigations of individual differences in children [51] and adults [64] and other researchers have often used the NDRT as a measure of reading ability [e.g.,13,14] both were included to appropriately assess the literature. In

addition, we wanted to further investigate findings from Lee et al. [74] which suggested that the comprehension subtests in these composites measure different aspects of reading ability.

Other tests included spelling [63], print exposure (Author Recognition [67]) and vocabulary knowledge (LexTALE [75]) which are good proxies for lexical quality (Lexical Quality Hypothesis [76–78]). The quality of an individual's lexical representations have been suggested to influence in the amount of information that can be extracted and used in parafoveal preview [24].

Significant differences in Rapid Automatized Naming (RAN) scores have previously been found between dyslexic and non-dyslexic adult readers by Kirkby et al. [47], who also found differences in the size of the transposed letter effect between these groups. Non-dyslexic adults performed significantly more quickly on alphanumeric RAN tasks than dyslexic adults and displayed larger differences between SL and TL previews in a similar reading experiment using an invisible boundary, therefore we included a measure of alphanumeric RAN [79] in the current test battery. Finally, a backwards digit span test was included as a measure of working memory capacity [80].

## Method

### Participants

Participants were 100 students and staff from the University of Southampton over the age of 18 (88 Females, mean age = 19.88 range = 18–40). Participants were all native English speakers with normal or corrected to normal vision and no known reading difficulties. Participants received course credits or £25. This research was reviewed and approved through the University of Southampton, Faculty of Environmental and Life Sciences Ethics Committee on 29th October 2021(ERGO Ref. 67732). Recruitment took place from 29/10/2021 to 10/06/2022.

### Apparatus

The sentences were presented on a 21" CRT monitor, with a refresh rate of 120 hz and a resolution of 1024 x 768, interfaced with a PC at a viewing distance of 60 cm. Sentences were presented in black, size 14, Courier New font on a grey background; three characters equated to approximately 1˚ of visual angle. Although reading was binocular, eye movements were recorded only from the right eye, using an EyeLink 1000 tracker (S.R. Research Ltd.), with forehead and chin rests in order to minimize head movements. The spatial resolution of the eye tracker was 0.05˚, and the sampling rate was 1000 hz.

Participants completed most of the tests and questionnaires during the study on a 14-inch Dell Laptop Computer. Such tests were administered via an online web browser running Qualtrics. Participants were required to respond using a variety of mouse responses and keyboard answers, and response times for timed elements were recorded via a timed mouse click integrated within Qualtrics. A computerised backwards digit span test was administered using Inquisit on a 19-inch DELL monitor (1024 × 768-pixel resolution). During Wechsler Individual Achievement Test (WIAT-II UK [72]) Reading Subtests researchers used the testing flip pad, scoring sheets and word/pseudoword cards included in the test pack.

### Materials

Sixty experimental sentences containing 6-letter target words were partially adapted from Pagán et al. [46]. Target words (nouns or adjectives) were bisyllabic with a CVC structure for the initial trigram, which was always within the same syllabic unit (e.g., monkey). Target words were embedded into neutral sentence frames and were rated on a scale of 0 (very unnatural to read) to 100 (very natural) (M = 75.37, SD = 8.48) by 18 participants who did not take

```
1. The blonde girl spotted the brown| monkey in the zoo. (ID)

2. The blonde girl spotted the brown| mnokey in the zoo. (TL)

3. The blonde girl spotted the brown| mrekey in the zoo. (SL)
```

**Fig 1. Example of an experimental sentence with three parafoveal preview conditions created for each target word.** A line represents the position of the invisible boundary for each sentence in this experiment. Once the reader's gaze crossed the invisible boundary (during a saccade) all previews were replaced with the correct target word.

part in the main experiment. Three preview conditions were generated for each target word; an identity (ID) condition, in which the preview of the target word was spelled correctly (e.g., monkey); a transposed letter (TL, e.g., mnokey) condition, where a preview was a nonword with the second and third letters transposed; or a substituted letter (SL, e.g., mrekey) condition, where the preview was a nonword with the second and the third letters substituted (see Fig 1). It has been noted that word-initial letters are especially important for word identification for both children and adults. White et al. [81] found that readers were more disrupted by transpositions of external letters (at the beginning e.g., rpoblem, or end e.g., problme, of a word) than internal letters (e.g., porblem/probelm) in a sentence reading study. They found that the greatest disruption to reading was seen in word-initial letter transpositions. For this reason, only internal letters were manipulated in the current experiment. Bigram frequencies for TL previews (M = 109.63, SD = 165.98) and SL previews (M = 101.25, SD = 150.93) were matched (t (118) = 0.77, p = .443) using the norms from the CELEX database [82]. None of the target transpositions or substitutions produced real words and all were orthographically illegal.

The three counterbalanced lists were presented within the eye tracking experiment. Participants were randomly assigned to one of these conditions and all read 5 practice sentences followed by 60 experimental sentences (20 per condition). The sentences occupied one line on the screen and the target always appeared in the middle of the sentence. Sentence order was randomised for each participant. Comprehension questions were included following 1/3 of the experimental sentences to encourage reading for comprehension.

Participants were asked two questions about their reading behaviour including "How often do you read for work?" and "How often do you read for leisure?". Next, participants' reading and cognitive skills were assessed by the following tests:

**Reading ability tests.** Participants completed the NDRT [73] which included a vocabulary task, where participants were asked to fill a blank space within a sentence with the most appropriate word. Single words were then presented, and participants were given multiple choices to select appropriate definitions. Next participants completed a reading comprehension task. Participants silently read passages presented on a screen, before answering comprehension questions that were presented below the passages (on the same screen). Participants were asked to record the line they had reached after 1 minute of reading on the first passage. The test was stopped after 10 minutes and answers were recorded for all questions they had answered in this time.

Participants also completed the WIAT-II [72]. This included a word reading subtest, where participants were asked to read a list of real words aloud from a sheet of paper which increased in difficulty. The experimenter marked participants' pronunciation accuracy and testing was stopped when the participant made six sequential errors. Next, participants were asked to read

a list of orthographically legal nonwords aloud from a sheet of paper (e.g., "flimp") in a pseudoword decoding subtest. The experimenter recorded the participants' pronunciation accuracy and testing was stopped after six sequential errors were made. Participants then completed the reading comprehension subtest, where they read passages (short fictional stories, informational text, advertisements, and how-to passages) aloud or silently before answering literal and inferential comprehension questions orally when asked by the experimenter. All subtests were combined to give an overall score for the whole test. Scores were normed according to test instructions and percentile scores were used in analyses.

**Vocabulary knowledge.**   To complete the LexTALE [75], participants were asked to indicate whether a word presented on screen was a real English word or a pseudoword. There was no time limit for this task.

**Spelling.**   Spelling [63] tests included spelling dictation and spelling recognition. Spelling dictation featured playback of 20 recorded key words. Participants then were asked to write down the correct word spellings. The words were also presented within sentences. Spelling recognition was made up of a list of 88 correctly and incorrectly spelled words. Participants had to select the incorrectly spelled words.

**Print exposure.**   In the Author Recognition Test [67], a list of real authors and foil names were presented, and participants were asked to identify which were the real ones. Participants were informed that the list featured some foil names.

**Rapid automatized naming.**   Alphanumeric RAN [79] tasks were used. A randomized series of letters or numbers in a 5x5 grid were presented onscreen. Time taken for participants to name the characters was recorded and the sum of the two conditions was used in analyses.

**Working memory capacity.**   Digits were presented in sequences of increasing lengths. Participants were asked to recall them first in the same order as they were presented, and then in backwards order. The length of the longest backwards sequence recalled correctly was recorded for each participant (Digit Span Backwards [80]).

**Design and procedure.**   Testing involved two sessions on different days. During the first session, participants read an information sheet and gave written consent, then completed the eye tracking task followed by experimenter administered WIAT-II and RAN tasks. For the eye tracking task, participants were asked to sit comfortably at the computer, resting their chin on a chinrest and were then guided through the set up and calibration of the eye tracker by the researcher. Participants were then required to direct their gaze to a fixation cross presented on the left of the screen. When ready, sentences were presented following the fixation cross. Participants were asked to read the sentences and answer questions presented on the screen using the keyboard to respond to ensure they were reading for comprehension. Participants could take breaks when needed.

During a second session participants took part in a separate eye tracking task (unrelated to the current study) and completed the reading comprehension and vocabulary subtests of the NDRT, LexTALE task, spelling dictation and spelling recognition tasks, Author Recognition test, RAN tasks and the backwards digit span task in a randomised order.

## Results

There were two stages of analysis. First, a principal components analysis was conducted to determine which cognitive tests shared variance and loaded together. Subsequently, factors extracted via PCA and tests that fell outside of these identified components were used to model eye movement measures (*first fixation durations* (FFD), *single fixation durations* (SFD), gaze durations (GD) and go past times).

## Individual differences tests

Overall WIAT-II scores were calculated using the age-adjusted scoring materials provided, which resulted in numerical scores as well as categorical assessments of reading proficiency (Borderline/ Low Average/ Average/ High Average/ Superior). Overall scores for the NDRT were calculated as an average of scores on the vocabulary and comprehension subtests. Overall spelling scores were calculated using an average of the spelling recognition and spelling dictation subtests [63]. Two participants were identified as having very high spelling dictation scores and extremely low spelling recognition scores (both identified almost every correct spelling rather than incorrect spelling) and it was considered highly likely that these participants had misread the instructions for the spelling recognition task given their otherwise high scores. As a result, spelling recognition scores for these two participants were reverse coded. The highest score on the backward version of the digit span was taken as an overall score [80]. For examining tests as single predictors of eye movements, in line with previous research, overall composite test scores were used. However, it is more appropriate to consider the cognitive skills that make up reading ability tests separately for PCA, subtests from the NDRT and WIAT-II reading ability tests were included from this point.

Participants met criteria for exclusion if their overall performance on multiple tasks was very poor, or if standardised reading assessments identified a potentially undiagnosed reading disorder. Potential outliers scoring very low on some tasks were assessed in terms of their performance on other tasks and were always found to score above average on standardised reading assessments (WIAT-II and NDRT) and therefore did not meet criteria for exclusion. Very high scores were retained since it is plausible to find individuals who are very high scoring on cognitive tasks within skilled reader populations. No participants were removed from the dataset and all 100 datasets were considered suitable for the current analyses. Descriptive statistics based on scores for each test are summarised In Table 1 below. Individual difference test scores were then centred to allow comparisons to be made between measures. Correlations for all test scores are presented in Table 2. Tests were moderately positively correlated, except for RAN and Digit span tests which were not (highly) correlated with other measures.

## Principal components analysis

A PCA was conducted using the in-built function 'prcomp' in R (version 4.2.2 [83]) to identify which tests loaded together to reduce the number of dimensions used in further analysis.

**Table 1. Descriptive statistics for tests and subtests (raw scores).**

|  | Min | Max | Mean | SD |
|---|---|---|---|---|
| NDRT Total | 71.00 | 147.00 | 118.66 | 16.59 |
| NDRT Comprehension | 22.00 | 74.00 | 57.28 | 9.94 |
| NDRT Vocabulary | 28.00 | 77.00 | 61.38 | 10.35 |
| WAIT Total | 84.00 | 134.00 | 115.73 | 9.98 |
| WIAT-II Comprehension | 71.00 | 124.00 | 109.73 | 10.00 |
| WIAT-II Pseudoword Decoding | 88.00 | 122.00 | 108.01 | 6.58 |
| WIAT-II Word Reading | 92.00 | 121.00 | 115.18 | 5.55 |
| Spelling | 13.00 | 63.00 | 37.54 | 11.28 |
| Author Recognition Test | -6.00 | 34.00 | 6.63 | 6.15 |
| LexTALE | 64.55 | 100.00 | 90.10 | 7.14 |
| Digit Span test | 3.27 | 8.75 | 5.87 | 1.20 |
| Rapid Automatized Naming | 28.79 | 63.66 | 39.87 | 7.39 |
| NDRT Words Per Minute | 80.00 | 610.00 | 251.43 | 85.10 |

**Table 2. Correlations between individual differences subtests.**

|  | NDRT Comp | NDRT Vocab | WIAT-II Comp | WIAT-II Pseudoword | WIAT-II Word Reading | Spelling | ART | LexTALE | Digit Span |
|---|---|---|---|---|---|---|---|---|---|
| NDRT Vocabulary | 0.34** | | | | | | | | |
| WIAT-II Comprehension | 0.15 | 0.60*** | | | | | | | |
| WIAT-II Pseudoword Decoding | 0.19 | 0.39*** | 0.36*** | | | | | | |
| WIAT-II Word Reading | 0.29** | 0.54*** | 0.47*** | 0.43*** | | | | | |
| Spelling | 0.31** | 0.60*** | 0.41*** | 0.55*** | 0.40*** | | | | |
| Author Recognition Test | 0.19 | 0.63*** | 0.36*** | 0.18 | 0.25* | 0.50*** | | | |
| LexTALE | 0.31** | 0.70*** | 0.47*** | 0.41*** | 0.45*** | 0.57*** | 0.53*** | | |
| Digit Span | 0.12 | 0.14 | 0.23* | 0.13 | -0.04 | 0.25* | 0.24* | 0.18 | |
| RAN | -0.07 | 0.09 | -0.02 | -0.03 | 0.01 | -0.07 | -0.15 | 0.00 | -0.13 |

Significance is denoted by

* $< .05$

** $< .01$

*** $< .001$.

NDRT and WIAT-II reading ability test scores are composite scores based on two or more subtests. For the remainder of this paper, subtests that comprise the WIAT-II (comprehension, word reading and pseudoword decoding) and NDRT (comprehension and vocabulary) were considered separately to assess each cognitive skill included within them. A single component was retained after a parallel analysis [84], this was conducted using the package 'paran' (version 1.5.2 [85]) in R (version 4.2.2 [83]). Parallel analysis calculates adjusted eigenvalues based on random noise expected using a simulated parallel dataset. According to this method, components that fall above the mean of the random eigenvalues should be retained. The component that met this criterion contributed 40.70% of variance in our data. Tests loadings on a component were considered important if contributions exceeded 10% (expected average contribution calculated from 1/number of variables = 1/10). In order of magnitude, the extracted component was explained by NDRT Vocabulary, LexTALE, Spelling, WIAT-II Comprehension, ART, and the word reading subtest of the WIAT-II (see Fig 2 below). We suggest that these tests are related in their assessment of lexical proficiency. The remaining tests fell outside of this component and were considered separately in subsequent analyses. We note that the comprehension subtests (NDRT and WIAT-II) were not found to load on the same component and will return to this in the discussion.

## Eye tracking analyses

Comprehension accuracy across trials was high (M = 98.82%, range = 85–100%), indicating that participants were reading for comprehension. Trials with blinks on the target word or featuring tracking loss were removed prior to analysis. Fixations shorter than 80 ms made within one character of a previous or subsequent fixation were merged and fixation durations outside of 80 ms to 800 ms were removed. Trials that involved fewer than 3 fixations across a sentence were also removed. Three trials were removed due to excessive blinking or tracker loss. Data were checked for instances where the boundary change was triggered by a saccade that subsequently landed on the preceding word, where a display changed occurred during a fixation on a pre-target word, or where the display changed had not completed until after 10 ms into the fixation on the target word. These trials were then removed (9.82%). Trials where the target word was skipped (3.06%) were also removed. Given the low skipping rates, word skipping

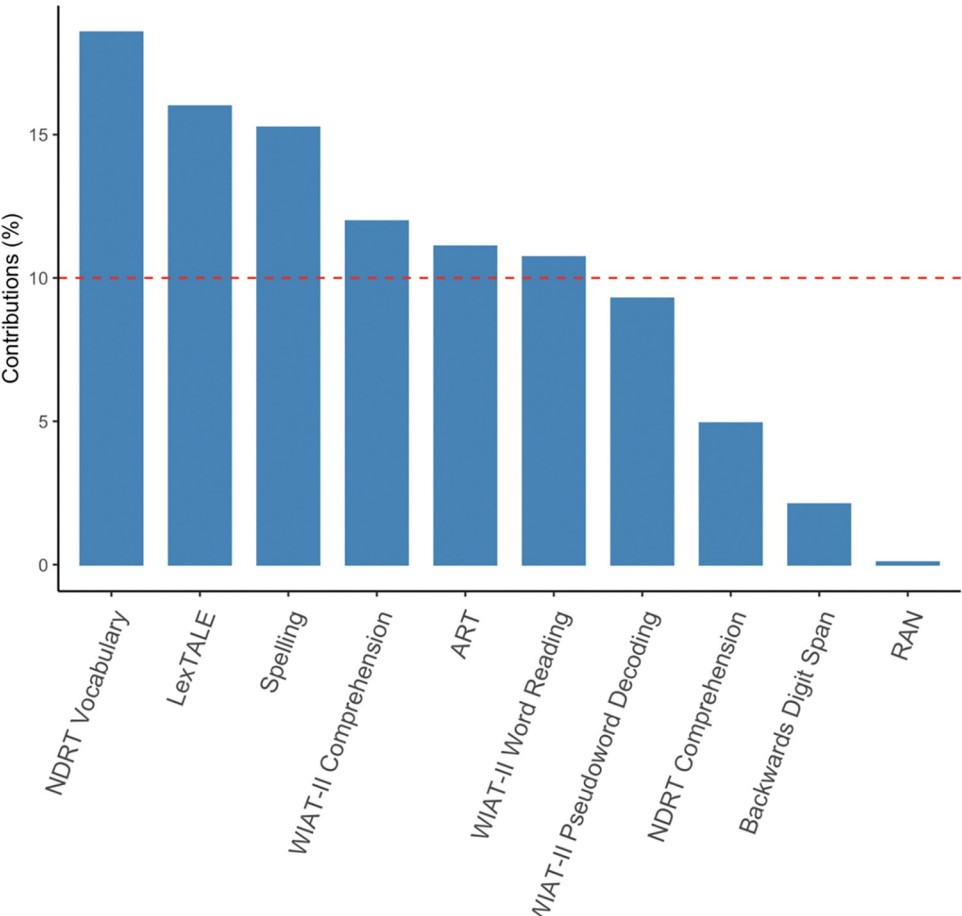

**Fig 2. Individual contributions of each individual differences measure on PC1.** The dotted line represents the expected average contribution (10%). A contribution above this line is considered important in explaining the component.

was not analysed. Outliers were then removed for each dependent variable; first fixation durations (FFD; the duration (ms) of the first fixation on a target word), single fixation durations (SFD; the duration (ms) of the fixation on a target word when it was only fixated exactly once), gaze durations (GD; the sum of all fixations (ms) made on a target word before moving from it in the first pass), and go past times (GOPAST; the sum of all fixations (ms) made on a target word before moving on to a later portion of the sentence, this includes regressions to previous portions of the sentence). Data falling outside of 2.5 standard deviations from the mean for each participant within a condition (high and low frequency) were removed (SFD; 1.44%, FFD; 1.68% GD; 1.91%, go past times; 1.72%). Descriptive statistics based on participant means for these measures are displayed in Table 3.

**Generalized linear mixed models.** All eye movement measures were analysed using the lme4 package (version 1.1–31 [86]) in R (version 4.2.2 [83]). For all Generalized Linear Mixed Models (GLMM), a Gamma distribution with identity link was used, following guidance for analysing skewed reaction time data without transformation [see 87] with participants and items as random factors.

Models were trimmed following the procedure described in Dirix and Duyck [88]. Models started with all fixed effects of interest: sliding difference contrasts for the orthographic preview conditions (TL-ID, and SL-TL), PC1, the NDRT comprehension, WIAT-II pseudoword

**Table 3. Descriptive statistics for eye movement measures.**

|  | Condition | Min | Max | Mean | SD |
|---|---|---|---|---|---|
| SFD | ID | 127.67 | 310.92 | 221.79 | 33.47 |
|  | TL | 154.67 | 330.17 | 229.53 | 38.05 |
|  | SL | 160.25 | 359.80 | 256.31 | 38.81 |
| FFD | ID | 149.25 | 306.73 | 220.46 | 32.47 |
|  | TL | 149.14 | 330.17 | 227.43 | 36.08 |
|  | SL | 160.25 | 349.15 | 246.04 | 33.53 |
| GD | ID | 159.67 | 402.53 | 250.24 | 47.60 |
|  | TL | 162.17 | 424.50 | 256.50 | 54.85 |
|  | SL | 160.25 | 448.64 | 290.20 | 52.07 |
| GOPAST | ID | 164.86 | 402.53 | 264.09 | 54.18 |
|  | TL | 165.86 | 454.38 | 270.76 | 60.84 |
|  | SL | 160.25 | 448.64 | 303.02 | 54.31 |

Means and SDs were calculated based on participant means per condition.

decoding, RAN and backwards digit span scores. Two-way interactions for each cognitive test and the orthographic preview conditions were also included. In addition, trial number and launch site were included as fixed factors to account for their potential influence on the data. Random factors were intercept only. This was the starting model. Next, non-significant interactions and afterwards fixed effects were sequentially checked to see if they could be removed without reducing model fit, starting with the effect with the largest p value. Model comparison Chi-square tests assessed whether removal of such effects impacted model fit. When models reached a point at which no further trimming was possible because all remaining fixed effects were significant, and therefore necessary to be retained, or if the removal of a non-significant effect would reduce model fit, we began building up the random effects structure to reach the maximal model (as is optimal for model analysis according to Barr et al. [89]). Random effects were forward fitted, with slopes for participant and item factors added. The order was as follows: orthographic preview condition, individual differences tests, launch site and finally trial number. Slopes were retained if the model converged and their addition improved the model fit. Finally, when the largest random structure was achieved, any non-significant fixed effects were again checked sequentially to see if trimming them would not reduce model fit.

Results, displayed in Tables 4 and 5, showed that there were small differences between ID and TL previews in FFDs (7.52 ms), SFDs (5.24 ms), GDs (8.16 ms, though this was only marginally significant) and go past times (10.02 ms). SL previews resulted in inflated FFD (17.24 ms), SFD (27.54 ms), GD (33.49 ms) and go past times (32.70 ms) on the target word compared to TL previews. When saccades were launched from positions close to the target, SFD, GD and go past times were shorter than when saccades were launched from a greater distance. Later trials in the experiment were associated with shorter SFDs, GDs and go past times compared to earlier trials, but no differences were found for FFDs on the target word. Participants who scored highly in tests associated with PC1 generally displayed shorter FFD, SFD, GD and go past times. However, these scores were not found to interact with any orthographic preview condition. No other main effects were found associated with individual differences measures. High scores on the backwards digit span task predicted significantly different SFDs for targets following TL previews compared to ID previews, however as shown in Fig 3, this interaction is completely encapsulated by 95% confidence intervals for ID and TL conditions on these measures, so we will not consider these further. No significant interactions were observed for any

**Table 4. GLMMS with multiple test predictors to predict first fixation durations and single fixation durations.**

| | First Fixation Duration | | | | Single Fixation Duration | | | |
|---|---|---|---|---|---|---|---|---|
| | Est | SE | t | p | Est | SE | t | p |
| (Intercept) | 233.59 | 3.89 | 59.99 | < .001 *** | 245.67 | 5.06 | 48.52 | < .001 *** |
| TL-ID | 7.52 | 3.74 | 2.01 | .044 * | 5.24 | 2.47 | 2.12 | .034* |
| SL-TL | 17.24 | 4.11 | 4.19 | < .001 *** | 27.54 | 2.65 | 10.38 | < .001 *** |
| Trial Number | - | - | - | - | 0.13 | 0.11 | -1.22 | .222 |
| Launch Site | - | - | - | - | 2.30 | 0.53 | 4.32 | < .001 *** |
| PC1 | 9.09 | 3.25 | 2.80 | .005 ** | 8.45 | 3.66 | 2.31 | .021 * |
| Backwards Digit Span | -2.61 | 3.38 | -0.77 | 0.43 | -2.14 | 3.54 | -0.60 | 0.55 |
| TL-ID*Backwards Digit Span | -5.51 | 3.05 | -1.81 | .071 | -5.75 | 2.45 | 2.35 | .019 * |
| SL-TL*Backward Digit Span | 2.90 | 3.48 | 0.83 | .405 | 0.99 | 2.54 | 0.39 | .696 |

Significance is denoted by

* < .05

** < .01

*** < .001. TL-ID represents the difference in times between the ID and TL conditions. SL-TL represents the difference between SL and TL conditions. The intercept refers to the grand mean.

other individual differences measures, letter position encoding flexibility remained stable across this population of skilled adult readers.

## Discussion

The present study examined whether the flexibility of letter position encoding reaches maturation in skilled adult reader populations or whether it varies in relation to differences in cognitive skills. We first grouped our battery of cognitive skills via overlapping variance by means of a PCA to reduce multicollinearity in models with multiple test predictors. We then analysed the transposed letter effect in relation to these cognitive skills on four eye movement measures (SFD, FFD, GD and go past times).

When saccades were launched from positions close to the target, SFDs, GDs and go past times on the target word were smaller than when saccades were launched from a greater distance. This is in line with previous research [e.g., 90] in that more information can be gathered from a

**Table 5. GLMMS with Multiple test predictors to predict gaze durations and go past times.**

| | Gaze Duration | | | | Go Past Times | | | |
|---|---|---|---|---|---|---|---|---|
| | Est | SE | t | p | Est | SE | t | p |
| (Intercept) | 277.38 | 5.68 | 48.85 | < .001 *** | 295.77 | 5.42 | 54.52 | < .001 *** |
| TL-ID | 8.16 | 4.33 | 1.89 | .059 . | 10.02 | 4.19 | 2.39 | .017 * |
| SL-TL | 33.49 | 4.79 | 6.99 | < .001 *** | 32.70 | 4.28 | 7.64 | < .001 *** |
| Trial Number | -0.17 | 0.08 | -2.24 | .025 * | -0.27 | 0.08 | -3.31 | .025 * |
| Launch Site | 6.58 | 0.63 | 10.49 | < .001 *** | 7.88 | 0.67 | 11.75 | < .001 *** |
| PC1 | 12.05 | 4.14 | 2.91 | .004 ** | 16.35 | 4.85 | 3.37 | < .001 *** |

Significance is denoted by

* < .05

** < .01

*** < .001. TL-ID represents the difference in times between the ID and TL conditions. SL-TL represents the difference between SL and TL conditions. The intercept refers to the grand mean.

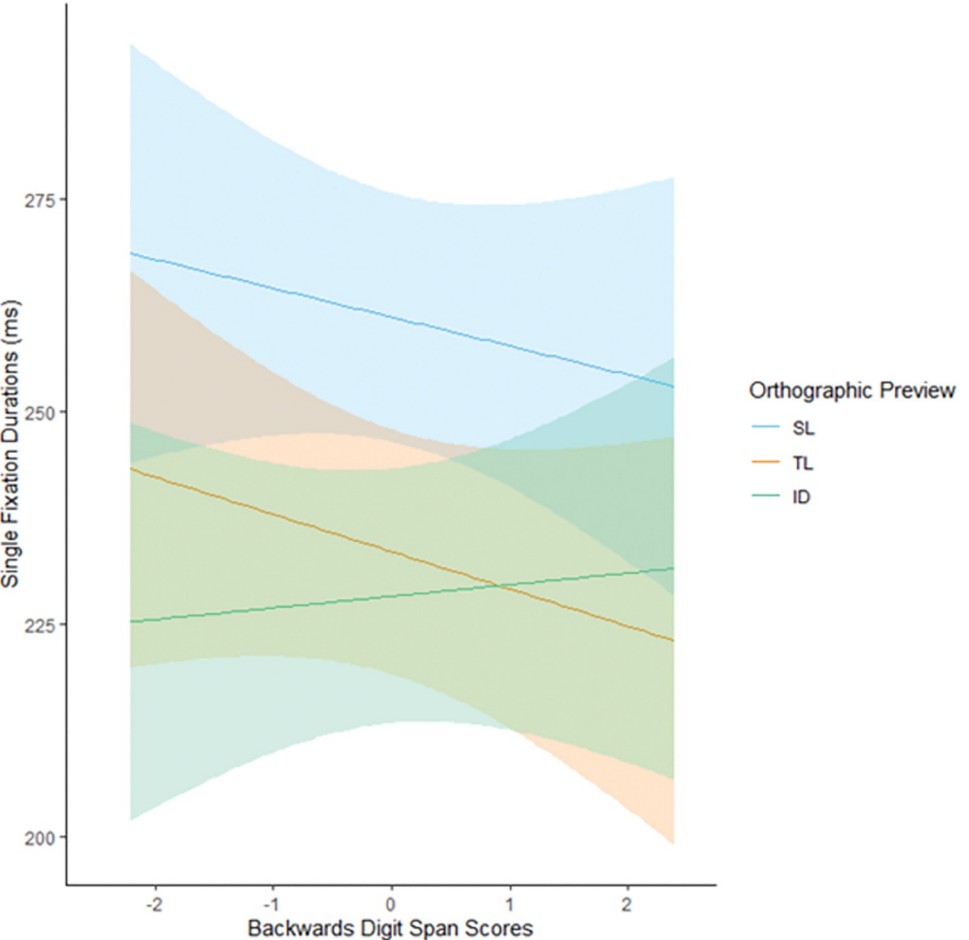

**Fig 3. The effect of orthographic preview on single fixation durations (ms) moderated by backwards digit span scores.** Shaded areas represent 95% confidence intervals.

parafoveal preview during fixations close to the upcoming word. As a result, less time is needed during a subsequent fixation on the upcoming word to identify it. Trials that occurred later in the experiment were associated with shorter SFDs, GDs and go past times than earlier trials indicating readers speeding up somewhat during the experiment, but no differences were found for FFDs.

## Transposed letter effect

Overall, transposed letter effects were found where SL previews resulted in increased fixation times on target words compared to TL previews. In comparison, there was only a small cost associated with a TL preview in comparison to an ID preview in FFDs, SFDs and go past times, with only a marginally significant difference in GDs, in line with previous findings in reading studies [43]. These findings are consistent with the idea that in general, skilled readers encode letter position more flexibly than letter identity.

## Individual differences

We adapted predictions based on the Multiple-route model [39] about developmental differences in letter position encoding to make some predictions about differences that may arise

within skilled adult readers. We noted that, if skilled readers generally rely on a coarse-grained route for orthographic decoding there may be a maturation of letter position encoding flexibility in skilled adult readers. This would result in all readers showing similar transposed letter effects with faster identification of target words following TL previews than SL previews. The alternative hypothesis was that only a subset of adult readers may rely more heavily on a coarse-grained route to orthographic decoding leading to larger differences between TL and SL preview conditions. If so, any differences in the transposed letter effect in adults might be observed in relation to individual differences in cognitive skills, as seen in developing readers [49,51].

We found no systematic differences in letter position encoding related to individual differences in cognitive skills in our main analyses, suggesting that the flexibility of letter position encoding in average-to-skilled adult readers remains fairly stable once reading has developed. At least this is the case for the adults in our sample who were classed as "average" to "superior" on the standardised WIAT-II reading ability. To fully investigate the compatibility of our results with previous literature about individual differences in parafoveal preview benefit [13,14,23,24] and differences in transposed letter effects in parafoveal preview in children's reading ability [49] we also ran analyses in which models only included single test predictors for spelling and reading ability. These analyses more closely reflect the analyses reported in these previous studies. The results of these analyses (available online at https://osf.io/b2rdm/?view_only=cf37e55f5c804a98bf7801a8c903d5f3) indicate that the only significant predictor of more flexible letter position encoding was the Nelson Denny reading test (reading comprehension and vocabulary composite score), where higher scores were associated with larger differences between SL and TL preview conditions. However, this observation was not stable across other eye movement measures as it was only observed in single fixation durations. Since this finding was restricted to a single measure and was not significantly predicted by another reading ability measure (WIAT-II reading test), our conclusion remains that the flexibility of letter position encoding in adult skilled readers reaches maturation (or near maturation) and varies very little in relation to cognitive skills.

We mentioned in the Introduction that parafoveal processing is a very early stage of word recognition, which takes place before a word is directly fixated. In order to establish that these null findings for individual differences in letter position encoding for skilled adults are not simply due to the limits of parafoveal processing, other paradigms should be investigated. For example, researchers may wish to determine whether variability of the transposed letter effect is seen during foveal processing, and whether such variability is predicted by individual skills. Future studies may utilise eye tracking with sentences containing misspelled words without a display change, similar to the paradigm used by Pagán et al., [49] when studying children's eye movements (we thank an anonymous reviewer for this suggestion).

Another way to extend this research would be to investigate the flexibility of letter position encoding in second language learners, who are simultaneously highly skilled L1 readers and developing L2 reader (we thank a second anonymous reviewer for this suggestion). Individual differences in reading skills in both languages may be included in such investigations, to address whether L1 and L2 reading skills (e.g., vocabulary size, spelling ability and reading experience) influence letter position encoding when reading in L2. It is worth noting here that a recent study [91] investigated transposed letter effects in native Chinese speakers learning English as a second language and found that L2 vocabulary knowledge (measured by the Lex-TALE) did not modulate the flexibility of readers' letter position encoding in English (L2).

We suggest that differences in the flexibility of letter position encoding in adults related to individual differences may appear where differences in cognitive skills are larger, for example where samples include struggling or developing readers. Hasenäcker and Schroeder [51] demonstrated that in a longitudinal study of children, the size of TL effect (in a lexical decision

task) was modulated by the readers' orthographic knowledge. We suspect that if there are important differences in the magnitude of the TL effect that occur in relation to reading skill in adults, these differences will be better seen across the entire population of poor to very skilled readers, as opposed to in our sample of average-to-very-skilled. Additionally, the target words in our sentences were not complex, and all skilled readers are likely to have been very successful at identifying them using a coarse-grained word reading strategy [39]. Future research may consider using more complex words to increase the power to discriminate between skilled readers in their use of phonological or orthographic word identification processes of less familiar words.

We stress that our findings are not in disagreement with previous research concerning real word primes. For example, in a masked priming experiment, Andrews and Lo [62] observed some modulation of the TL effect related to individual differences when primes were real words (e.g., salt/slat). However, crucially, Andrews and Lo did not find any modulation of the TL effect for pseudoword primes similar to those used in the present experiment. Here we must highlight an important difference between word and pseudoword primes. Pseudoword previews that are perceived to be similar to the target word facilitate recognition of the target word due to activation of similar orthographic features without competing lexical information (because they are not real words). Whereas, if the preview is a visually similar real word, it will instead cause a delay when processing the target word due to the additional requirement of diverting resources away from lexical retrieval of the preview word. The lexical information associated with the preview word will inhibit rapid retrieval of lexical information associated with the target word, which overrides the facilitation of overlapping orthographic features [92].

Similar to our previous investigations of individual differences in skilled adult readers [74], a PCA in the current analyses grouped together skills that have previously been linked to lexical quality (vocabulary, spelling ability and reading experience [76–78,93]. We found that these skills were related to general sentence processing with faster fixations and shorter reading times associated with high levels of reading skill, in line with previous research [74]. This is consistent with the Lexical Quality Hypothesis [76–78] in that words were identified more quickly by participants with higher quality lexical representations indexed by high scores on this component, and is in line with previous research about skills associated with lexical quality [94,95].

Importantly, we note that in our PCA, the comprehension subtest of the WIAT-II was associated with lexical proficiency (PC1) whereas the comprehension subtest of the NDRT was distinct and that these tests were weakly correlated ($r = 0.15$). Although both are standardised measures of comprehension, it appears that they measure distinct underlying constructs, (as also noted in a previous investigation [74]). Previous research has suggested that NDRT comprehension scores are associated with IQ [96–97]. Future research may include measures of IQ within a test battery to investigate these ideas further. Differences in constructs measured by tests that share a name provide an example of Thorndike's *Jingle fallacy* [98]. Therefore, reading comprehension measures should be selected carefully for future research.

## Conclusion

We conclude that, in general, skilled adults encode letter position more flexibly than letter identity, with a greater processing cost associated with changes in letter identity than changes in letter position. We observed very few individual differences in the flexibility of skilled readers' letter position encoding, suggesting that letter position encoding reaches maturation (or near maturation) and is fairly stable for skilled adult readers.

## Acknowledgments

Special thanks to Karolina Vakulya for her work on data collection for this research.

## Author Contributions

**Conceptualization:** Charlotte E. Lee, Ascensión Pagán, Denis Drieghe.

**Data curation:** Charlotte E. Lee, Denis Drieghe.

**Formal analysis:** Charlotte E. Lee, Denis Drieghe.

**Funding acquisition:** Charlotte E. Lee, Denis Drieghe.

**Investigation:** Charlotte E. Lee, Denis Drieghe.

**Methodology:** Charlotte E. Lee, Ascensión Pagán, Denis Drieghe.

**Project administration:** Charlotte E. Lee, Denis Drieghe.

**Resources:** Charlotte E. Lee, Denis Drieghe.

**Software:** Charlotte E. Lee, Denis Drieghe.

**Supervision:** Denis Drieghe.

**Validation:** Charlotte E. Lee, Denis Drieghe.

**Visualization:** Charlotte E. Lee, Denis Drieghe.

**Writing – original draft:** Charlotte E. Lee, Ascensión Pagán, Hayward J. Godwin, Denis Drieghe.

**Writing – review & editing:** Charlotte E. Lee, Ascensión Pagán, Hayward J. Godwin, Denis Drieghe.

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
