## [Decision Letter · Decision Letter 0]

30 Oct 2023

PONE-D-23-23508Individual differences and the transposed letter effect during readingPLOS ONE

Dear Dr. Lee,

Thank you very much for submitting your manuscript to Plos One. I have received the comments of two expert reviewers on visual word recognition. I thank them for their careful reading and informed comments. Based on their analyses and on my own evaluation, I recommend a major review. Below you can find the reviewers´ suggestions. We invite you to submit a revised version of the manuscript that addresses the points raised during the review process.

We look forward to receiving your revised manuscript.

Kind regards,

Montserrat Comesaña Vila

Academic Editor

PLOS ONE

2. We note that you have referenced (34. Davis CJ. The self-organising lexical acquisition and recognition (SOLAR) model of visual word recognition. Unpublished doctoral dissertation, University of New South Wales, Australia; 1999. doi:10.26190/unsworks/13769) which has currently not yet been accepted for publication. Please remove this from your References and amend this to state in the body of your manuscript: (ie “Bewick et al. [Unpublished]”) as detailed online in our guide for authors

Reviewers' comments:

Reviewer's Responses to Questions

**Comments to the Author**

1. Is the manuscript technically sound, and do the data support the conclusions?

Reviewer #1: Yes

Reviewer #2: Yes

2. Has the statistical analysis been performed appropriately and rigorously? 

Reviewer #1: Yes

Reviewer #2: Yes

3. Have the authors made all data underlying the findings in their manuscript fully available?

Reviewer #1: Yes

Reviewer #2: Yes

4. Is the manuscript presented in an intelligible fashion and written in standard English?

Reviewer #1: Yes

Reviewer #2: Yes

5. Review Comments to the Author

Reviewer #1: The aim of the study is to investigate individual differences in parafoveal processing of upcoming words during reading, and to determine whether such differences may be related to individual differences in in reading ability of the participants. This was achieved by evaluating adult participants using a sentence reading task with an invisible boundary paradigm, as per Pagán et al., 2016.

The introduction provides a good summary of current ideas regarding the structure/organisation of orthographic information within the lexicon, specifically, regarding the how the flexibility of letter identity coding and letter position coding change with age. The authors cite various studies which have shown that children’s reading ability moderates the effects of Transposed Letter (TL) priming. Hence, it is currently believed that flexibility in letter position coding increases with age, but is moderated by “reading skill” (there is heterogeneity in the literature in terms of what specific aspect of reading skill moderates the TL effect). In support of this idea the authors describe Multiple-route model (Grainger & Ziegler, 2011). However, the possible role if individual differences is understudied in adults, and it appears nobody has previously explored this idea using TL priming. Thus, the authors wondered if individual differences in reading related skills in adults might also moderate TL priming effects. To explore this issue the authors used a sentence reading task with an invisible boundary paradigm, TL priming, and assessed participants using a large battery of cognitive tasks.

I enjoyed reading this manuscript and think it merits publication. Below I make some comments and suggestions which I think should be relatively simple for the authors to deal with.

Method

Although it is fairly easy to find more details about the design from reading Pagán et al 2016, I think the manuscript would benefit by having a figure similar to Figure 1 in that article to help the reading better understand the sentence reading task.

Results

The use of PCA analysis to reduce the number of predictors is sensible, although perhaps the authors could provide a little bit of detail as to how this was carried out; for example in R, and if so, which packages? If not R, what platform/tool was used?

The treatment of outliers, both in terms of entire participant and individual trials, along with the cleaning of the eye movement data is both clear and reasonable.

I’m a little confused about the presentation of results in Tables 4 and 5. Presumably, the intercept represents the reference category of the identity condition. Hence the “TL-ID” line in the table seems to represent the difference in times between the LD and TL conditions (and this is supported by the text on P23 L75. L76-78 then talk about comparing the SL and TL conditions, implying that the “SL-TL” line in the tables represents this difference. However, unless I am mistaken, this could only be done of the reference category was changed (which, of course, you have to do), although no mention was made of this. Could the authors please clarify this point?

I appreciated the additional analysis in the supplementary materials despite the fact that they were largely “null results”.

However, I found that the discussion/conclusion ended rather abruptly. I understand that the authors would have had more “material” to talk about had individuals differences turned out to moderate the effects. However, the authors perhaps could talk about possible future extensions to this work. For example, it seems to me that evaluating adults learning a second language might be an interesting avenue of investigation given the combination of a fully developed brain but basic 2L knowledge/skill.

Minor

P29 L18. Perhaps better with a couple of commas; “We conclude that, in general, skilled adults”

P13 L 149. “but that letter position encoding for words is already fairly flexible in early reading development. These investigations demonstrate that letter position encoding becomes more flexible as reading skills improve”. I agree that letter position coding probably does become more flexible as reading improves (as per Grainger & Ziegler, 2011). However, these two sentences seem somewhat contradictory. Possibly it’s worth revising the wording.

P16. L204 should “larger” be “large”?

P19. L 281. What database was used to calculate the bigram frequencies?

Grainger J, Ziegler JC. A dual-route approach to orthographic processing. Front Psychol. 2011;2:54. doi:10.3389/fpsyg.2011.00054

Pagán A, Blythe HI, Liversedge SP. Parafoveal preprocessing of word initial trigrams during reading in adults and children. J Exp Psychol Learn Mem Cogn. 2016;42(3):411-432. doi:10.1037/xlm0000175

Reviewer #2: Review of ms “Individual differences and the transposed letter effect during reading”

This manuscript presents an eye movement experiment during reading that manipulates the relationships between the parafoveal previews (identity, transposed-letter, replacement-letter) and the target words, and examines whether the transposition effects (TL vs RL, and also ID vs. TL) are modulated by a range of measures related to individual differences.

The results were very clear. There was a strong TL vs. RL difference in the parafovea, whereas the ID vs. TL difference was much weaker, as is usual in previous research on the topic starting with Johnson et al. (2007, JEP:LMC). At the same time, there was essentially no evidence for the role of the predictors related individual differences as modulators of the above parafoveal effects.

As Jeff Rouder and others have often stressed, invariances offer valuable information, and here what we see is that individual differences do not shape the transposed letter effect when using parafoveal previews, which is interesting by itself. So I am quite positive, and I think that, pending some revision, this manuscript should be published. I would also like to add that the structure of the paper and writing quality were excellent and I enjoyed reading the manuscript.

Here are some issues that the authors may want to tackle in a revised version of the manuscript:

—The authors should differentiate more clearly between those letter transposition effects resulting from normal reading (or from single presentation techniques) and those come from parafoveal previews (or from masked priming). This is a point first raised a long time ago by Andrews (1997, PBR) and has been echoed a number of times since. In a parafoveal preview experiment, it’s the (rapid) integration between an explicitly presented preview and the target what matters, let alone that these effects originate very early during letter and word encoding. In a “normal” reading scenario, there is no explicit activation from a preview, so the transposed-letter effects are instead inferred from the manipulation.

—The presence of individual differences effects in transposed-letter effects during normal reading without parafoveal manipulations (or with, say, single-presentations lexical decision tasks) reported in previous experiments is not at odds with a lack of such modulation in the parafovea shown here. These are just separate research questions. I don’t recall now the literature on parafoveal previews and identity effects from a developmental perspective, but in masked priming, the pattern of identity priming effects in Grade 2 children, Grade 4 children, and adult readers is quite similar, including the presence of a shift of the latency distributions (e.g., Gomez & Perea, 2020, JECP). As the reading skills of Grade 2 children are necessarily lower than those of Grade 4 children (and those of adults), what this means is that letter encoding (at least in the fovea) was quite efficient even for young reader, thus leaving little room for a modulation of individual differences in the very first moments of processing. In the paper by Andrews and Lo (2012, JEPLMC), the (small) modulations of masked priming effects due to individual differences occurred for “word primes” (salt-SLAT), but not for the type of pseudoword primes that have been used in the present experiment. That is, the present findings are convergent with the Andrews and Lo paper—it’s just that the present experiment did not use word (TL/SL) previews.

—The Discussion section needs to be slightly rewritten to wrap up the apparent divergences with previous research, which are just apparent but they have an easy explanation. Here there is no modulation of the transposed letter effect as a function of the subjects’ reading skills because these early effects are very difficult to capture in the first place. This modulation would have been easier to catch using a paradigm like that of Rayner et al. (2016, PBR) with jumbled words in the middle of a sentence, and this issue can be left for future work. Based on previous work from Pagán et al. (2021, JEPLMC) and from Perea et al. (2021, QJEP), I would expect the modulation of TL effects as a function of individual differences in a sentence reading experiment without parafoveal previews.

6. PLOS authors have the option to publish the peer review history of their article (what does this mean?). If published, this will include your full peer review and any attached files.

Reviewer #1: No

Reviewer #2: No

---

## [Author Response · Author response to Decision Letter 0]

11 Dec 2023

Dear Prof. Comesaña Vila

First, my co-authors and I would like to thank you and all the reviewers for the very helpful comments on the previous version of this paper. 

We believe we have now successfully addressed the issues raised, and have indicated where these changes have been made both in the manuscript (highlighted) and in the text below.

Kind regards, 

Charlotte Lee

2. We note that you have referenced (34. Davis CJ. The self-organising lexical acquisition and recognition (SOLAR) model of visual word recognition. Unpublished doctoral dissertation, University of New South Wales, Australia; 1999. doi:10.26190/unsworks/13769) which has currently not yet been accepted for publication. Please remove this from your References and amend this to state in the body of your manuscript: (ie “Bewick et al. [Unpublished]”) as detailed online in our guide for authors

This reference has been updated to cite a published paper by C.J. Davis for this model (reference on page 46, line 833).

Davis, CJ. The spatial coding model of visual word identification. Psychological review. 2010; 117(3), 713. doi:10.1037/a0019738

Review Comments to the Author

Reviewer #1: The aim of the study is to investigate individual differences in parafoveal processing of upcoming words during reading, and to determine whether such differences may be related to individual differences in in reading ability of the participants. This was achieved by evaluating adult participants using a sentence reading task with an invisible boundary paradigm, as per Pagán et al., 2016.

The introduction provides a good summary of current ideas regarding the structure/organisation of orthographic information within the lexicon, specifically, regarding the how the flexibility of letter identity coding and letter position coding change with age. The authors cite various studies which have shown that children’s reading ability moderates the effects of Transposed Letter (TL) priming. Hence, it is currently believed that flexibility in letter position coding increases with age, but is moderated by “reading skill” (there is heterogeneity in the literature in terms of what specific aspect of reading skill moderates the TL effect). In support of this idea the authors describe Multiple-route model (Grainger & Ziegler, 2011). However, the possible role if individual differences is understudied in adults, and it appears nobody has previously explored this idea using TL priming. Thus, the authors wondered if individual differences in reading related skills in adults might also moderate TL priming effects. To explore this issue the authors used a sentence reading task with an invisible boundary paradigm, TL priming, and assessed participants using a large battery of cognitive tasks.

I enjoyed reading this manuscript and think it merits publication. Below I make some comments and suggestions which I think should be relatively simple for the authors to deal with.

Response: We thank the reviewer for their very positive evaluation of our manuscript. 

Reviewer 1: Method

Although it is fairly easy to find more details about the design from reading Pagán et al 2016, I think the manuscript would benefit by having a figure similar to Figure 1 in that article to help the reading better understand the sentence reading task.

Response: We thank the reviewer for this suggestion and have now added a figure to demonstrate the conditions more clearly in the stimuli section on page 18 line 345. We have also now included a file containing the stimuli and bigram frequencies of each preview word on OSF.

R1: Results

The use of PCA analysis to reduce the number of predictors is sensible, although perhaps the authors could provide a little bit of detail as to how this was carried out; for example in R, and if so, which packages? If not R, what platform/tool was used?

Response: PCA was calculated in R using the in-built function prcomp. Parallel analyses were conducted using the package paran in R. We have now added this information manuscript on page 27 lines 476 and 483 with relevant citations and version numbers.

R1: The treatment of outliers, both in terms of entire participant and individual trials, along with the cleaning of the eye movement data is both clear and reasonable.

I’m a little confused about the presentation of results in Tables 4 and 5. Presumably, the intercept represents the reference category of the identity condition. Hence the “TL-ID” line in the table seems to represent the difference in times between the LD and TL conditions (and this is supported by the text on P23 L75. L76-78 then talk about comparing the SL and TL conditions, implying that the “SL-TL” line in the tables represents this difference. However, unless I am mistaken, this could only be done of the reference category was changed (which, of course, you have to do), although no mention was made of this. Could the authors please clarify this point?

Response: We thank the reviewer for pointing out how we could make the description of our analyses more transparent. Sliding differences contrasts were used in our analyses, such that TL-ID represents the difference in times between the ID and TL conditions, and SL-TL represents the difference between SL and TL conditions in the tables and the intercept refers to the grand mean. We have now added a note explaining the contrasts below the tables and we have also clarified this in the description of the model structure on page 30 line 539.

R1: I appreciated the additional analysis in the supplementary materials despite the fact that they were largely “null results”.

However, I found that the discussion/conclusion ended rather abruptly. I understand that the authors would have had more “material” to talk about had individuals differences turned out to moderate the effects. However, the authors perhaps could talk about possible future extensions to this work. For example, it seems to me that evaluating adults learning a second language might be an interesting avenue of investigation given the combination of a fully developed brain but basic 2L knowledge/skill.

Response: We thank reviewer #1 for this suggestion that we have now discussed on page 38 line 669and also reviewer #2 for another suggestion for future research that we now discuss on page 37 line 658. We acknowledge in the paper that these suggestions came form our reviewers.

R1: Minor

P29 L18. Perhaps better with a couple of commas; “We conclude that, in general, skilled adults”

Response: We are happy to add clearer punctuation here on page 41 line 731.

R1: P13 L 149. “but that letter position encoding for words is already fairly flexible in early reading development. These investigations demonstrate that letter position encoding becomes more flexible as reading skills improve”. I agree that letter position coding probably does become more flexible as reading improves (as per Grainger & Ziegler, 2011). However, these two sentences seem somewhat contradictory. Possibly it’s worth revising the wording.

Response: We thank the reviewer for pointing out some confusing wording here. This section has now been revised on page 10 lines 186-190. It now reads: “They concluded that whilst letter position encoding for words is fairly flexible in early readers, it becomes more flexible as reading skills are developed further. They suggested that such changes are driven by increasing orthographic knowledge in children, for which grade is a good proxy.”

R1: P16. L204 should “larger” be “large”?

Response: This typo has now been corrected on page 13 line 251.

R1:P19. L 281. What database was used to calculate the bigram frequencies?

Response: Bigram frequencies were calculated using the CELEX database (Baayen, Piepenbrock, & Gulikers, 1995), we have now added this information on page 18 line 342.

Grainger J, Ziegler JC. A dual-route approach to orthographic processing. Front Psychol. 2011;2:54. doi:10.3389/fpsyg.2011.00054

Pagán A, Blythe HI, Liversedge SP. Parafoveal preprocessing of word initial trigrams during reading in adults and children. J Exp Psychol Learn Mem Cogn. 2016;42(3):411-432. doi:10.1037/xlm0000175

Reviewer #2: Review of ms “Individual differences and the transposed letter effect during reading”

This manuscript presents an eye movement experiment during reading that manipulates the relationships between the parafoveal previews (identity, transposed-letter, replacement-letter) and the target words, and examines whether the transposition effects (TL vs RL, and also ID vs. TL) are modulated by a range of measures related to individual differences.

The results were very clear. There was a strong TL vs. RL difference in the parafovea, whereas the ID vs. TL difference was much weaker, as is usual in previous research on the topic starting with Johnson et al. (2007, JEP:LMC). At the same time, there was essentially no evidence for the role of the predictors related individual differences as modulators of the above parafoveal effects.

As Jeff Rouder and others have often stressed, invariances offer valuable information, and here what we see is that individual differences do not shape the transposed letter effect when using parafoveal previews, which is interesting by itself. So I am quite positive, and I think that, pending some revision, this manuscript should be published. I would also like to add that the structure of the paper and writing quality were excellent and I enjoyed reading the manuscript.

Response: We also thank this reviewer for their very positive evaluation of the manuscript. 

R2: Here are some issues that the authors may want to tackle in a revised version of the manuscript:

—The authors should differentiate more clearly between those letter transposition effects resulting from normal reading (or from single presentation techniques) and those come from parafoveal previews (or from masked priming). This is a point first raised a long time ago by Andrews (1997, PBR) and has been echoed a number of times since. In a parafoveal preview experiment, it’s the (rapid) integration between an explicitly presented preview and the target what matters, let alone that these effects originate very early during letter and word encoding. In a “normal” reading scenario, there is no explicit activation from a preview, so the transposed-letter effects are instead inferred from the manipulation. 

—The presence of individual differences effects in transposed-letter effects during normal reading without parafoveal manipulations (or with, say, single-presentations lexical decision tasks) reported in previous experiments is not at odds with a lack of such modulation in the parafovea shown here. These are just separate research questions. I don’t recall now the literature on parafoveal previews and identity effects from a developmental perspective, but in masked priming, the pattern of identity priming effects in Grade 2 children, Grade 4 children, and adult readers is quite similar, including the presence of a shift of the latency distributions (e.g., Gomez & Perea, 2020, JECP). As the reading skills of Grade 2 children are necessarily lower than those of Grade 4 children (and those of adults), what this means is that letter encoding (at least in the fovea) was quite efficient even for young reader, thus leaving little room for a modulation of individual differences in the very first moments of processing. In the paper by Andrews and Lo (2012, JEPLMC), the (small) modulations of masked priming effects due to individual differences occurred for “word primes” (salt-SLAT), but not for the type of pseudoword primes that have been used in the present experiment. That is, the present findings are convergent with the Andrews and Lo paper—it’s just that the present experiment did not use word (TL/SL) previews. 

—The Discussion section needs to be slightly rewritten to wrap up the apparent divergences with previous research, which are just apparent but they have an easy explanation. Here there is no modulation of the transposed letter effect as a function of the subjects’ reading skills because these early effects are very difficult to capture in the first place. This modulation would have been easier to catch using a paradigm like that of Rayner et al. (2016, PBR) with jumbled words in the middle of a sentence, and this issue can be left for future work. Based on previous work from Pagán et al. (2021, JEPLMC) and from Perea et al. (2021, QJEP), I would expect the modulation of TL effects as a function of individual differences in a sentence reading experiment without parafoveal previews.

Response: We thank the reviewer for this detailed suggestion. We have now included some clarification and discussion of these points in the introduction and discussion sections highlighted in the manuscript:

Pages 6-7 lines 111-130

Page 8 lines 139-150

Page 37 lines 658-668

Pages 39-40 lines 695 - 710

---

## [Editor Report · Decision Letter 1]

28 Dec 2023

PONE-D-23-23508R1Individual differences and the transposed letter effect during readingPLOS ONE

Dear Dr. Lee, Thank you for submitting your manuscript to PLOS ONE. After careful consideration, we feel that it has merit but does not fully meet PLOS ONE’s publication criteria as it currently stands. Therefore, we invite you to submit a revised version of the manuscript that addresses the points raised during the review process. Please submit your revised manuscript by Feb 11 2024 11:59PM. If you will need more time than this to complete your revisions, please reply to this message or contact the journal office at plosone@plos.org. Please include the following items when submitting your revised manuscript:A rebuttal letter that responds to each point raised by the academic editor and reviewer(s). You should upload this letter as a separate file labeled 'Response to Reviewers'.A marked-up copy of your manuscript that highlights changes made to the original version. You should upload this as a separate file labeled 'Revised Manuscript with Track Changes'.An unmarked version of your revised paper without tracked changes. You should upload this as a separate file labeled 'Manuscript'.If applicable, we recommend that you deposit your laboratory protocols in protocols.io to enhance the reproducibility of your results. Protocols.io assigns your protocol its own identifier (DOI) so that it can be cited independently in the future. For instructions see: https://journals.plos.org/plosone/s/submission-guidelines#loc-laboratory-protocols. Additionally, PLOS ONE offers an option for publishing peer-reviewed Lab Protocol articles, which describe protocols hosted on protocols.io. Read more information on sharing protocols at https://plos.org/protocols?utm_medium=editorial-email&utm_source=authorletters&utm_campaign=protocols.

We look forward to receiving your revised manuscript.

Kind regards,

Montserrat Comesaña Vila

Academic Editor

PLOS ONE

Journal Requirements:

**Additional Editor Comments:**

Thank you again for submitting your manuscript ID PONE-D-23-23508R1 entitled "Individual differences and the transposed letter effect during reading " to Plos One. Your manuscript has been reviewed again by myself and i truly think you have adressed the suggestions made by the two reviewers and improved the manuscript. Thus, I am very happy to accept it with minor revision. Below you could find my suggestions.

In the abstract please make explicit what the acronym PCA stands for (Principal Component Analysis)

Line 139, please replace “Reading studies” with “Sentence-reading studies”

Line 168, remove the comma in Gómez et al., [50]

Lines 173-175, you stated that lexical decision times were modulated by individual differences in reading ability but then you say that “negligible differences were associated with word-reading”. When you say “associated with word-reading” did you mean associated with the RTs found in the lexical decision task or using other tasks with other non-experimental words? Re-write this in order to be clearer.

Line 177, when you talk about the principal componente analysis here put the acronym between parentheses.

Line 283 please makes explicit what the acronym RAN stands for (Rapid Automatized Naming)

Line 295 an space after the = sign is needed

Lines 336-338 please review the examples you gave because there are no transposed letters at all except for the internal letters:

(at the beginning e.g.,problem, or end e.g., problem, of a word) than internal letters (e.g., problem/probelm) in a sentence reading study.

Lines 668-679. Please include what i have written below in uppercase.

“…used by Pagán et al., [49] when studying children’s eye movements (we thank an anonymous reviewer for this suggestion). Another way to extend this research would be to investigate the flexibility of letter position encoding in second language learners, who are simultaneously highly skilled L1 readers and developing L2 reader (we thank a second anonymous reviewer for this suggestion). Individual differences in reading

skills in both languages may be included in such investigations, to address

whether L1 and L2 reading skills (e.g., vocabulary size, spelling ability and

reading experience) influence letter position encoding when reading in L2. IT IS WORTH NOTING HERE THAT a recent study [91] investigated transposed letter effects in native Chinese speakers learning English as a second language and found that L2 vocabulary knowledge (measured by the LexTALE) did not modulate the flexibility of readers’ letter position encoding in English (L2).”

Lines 721-725, you stated that “the comprehension subtest of the

WIAT-II was associated with lexical proficiency (PC1) whereas the

comprehension subtest of the NDRT was distinct and that these tests were

weakly correlated (r = 0.15). Although both are standardised measures of

comprehension, it appears that they measure distinct underlying constructs”.

Could you speculate about the construct thai is measured by the NDRT?

---

## [Author Response · Author response to Decision Letter 1]

19 Jan 2024

Dear Prof. Comesaña Vila,

My co-authors and I would like to thank you for your comments on previous versions of this manuscript. We were happy to make the changes that you have suggested. Below we provide some additional information to address the comments. We have also highlighted changes made in the revised manuscript.

Kind regards, 

Charlotte Lee

PONE-D-23-23508R1

Individual differences and the transposed letter effect during reading

PLOS ONE

Editor Comments: 

Thank you again for submitting your manuscript ID PONE-D-23-23508R1 entitled "Individual differences and the transposed letter effect during reading " to Plos One. Your manuscript has been reviewed again by myself and i truly think you have adressed the suggestions made by the two reviewers and improved the manuscript. Thus, I am very happy to accept it with minor revision. Below you could find my suggestions.

In the abstract please make explicit what the acronym PCA stands for (Principal Component Analysis)

Response: We have now clarified that the acronym PCA refers to a principal components analysis in the abstract on line 33-34.

EC: Line 139, please replace “Reading studies” with “Sentence-reading studies”

Response: We have now made this change on line 140.

EC: Line 168, remove the comma in Gómez et al., [50]

Response: This has now been removed on line 169.

EC: Lines 173-175, you stated that lexical decision times were modulated by individual differences in reading ability but then you say that “negligible differences were associated with word-reading”. When you say “associated with word-reading” did you mean associated with the RTs found in the lexical decision task or using other tasks with other non-experimental words? Re-write this in order to be clearer.

Response: We have now included clarification on lines 174-175 that word reading skill was measured by a subtest from PROLEC-R [Error! Reference source not found.], rather than during the lexical decision task. 

EC: Line 177, when you talk about the principal componente analysis here put the acronym between parentheses.

Response: The acronym (PCA) has now been included on line 179.

EC: Line 283 please makes explicit what the acronym RAN stands for (Rapid Automatized Naming)

Response: This now reads “Rapid Automatized Naming (RAN) scores” on line 284.

EC: Line 295 an space after the = sign is needed

Response: This has now been added on line 296.

EC: Lines 336-338 please review the examples you gave because there are no transposed letters at all except for the internal letters:

(at the beginning e.g.,problem, or end e.g., problem, of a word) than internal letters (e.g., problem/probelm) in a sentence reading study. 

Response: The relevant letter transpositions have now been corrected in the examples on lines 338 and 339. We thank you for highlighting this issue. 

EC: Lines 668-679. Please include what i have written below in uppercase.

 “…used by Pagán et al., [49] when studying children’s eye movements (we thank an anonymous reviewer for this suggestion). Another way to extend this research would be to investigate the flexibility of letter position encoding in second language learners, who are simultaneously highly skilled L1 readers and developing L2 reader (we thank a second anonymous reviewer for this suggestion). Individual differences in reading

skills in both languages may be included in such investigations, to address

whether L1 and L2 reading skills (e.g., vocabulary size, spelling ability and

reading experience) influence letter position encoding when reading in L2. IT IS WORTH NOTING HERE THAT a recent study [91] investigated transposed letter effects in native Chinese speakers learning English as a second language and found that L2 vocabulary knowledge (measured by the LexTALE) did not modulate the flexibility of readers’ letter position encoding in English (L2).”

Response: This wording has now been included on lines 676 – 677.

EC: Lines 721-725, you stated that “the comprehension subtest of the

WIAT-II was associated with lexical proficiency (PC1) whereas the

comprehension subtest of the NDRT was distinct and that these tests were

weakly correlated (r = 0.15). Although both are standardised measures of

comprehension, it appears that they measure distinct underlying constructs”.

Could you speculate about the construct thai is measured by the NDRT? 

Response: Thank you for this comment. Although we are a bit hesitant to speculate on the construct measured by the NDRT comprehension test from our data, we have now provided a suggestion (on lines 727 – 730) from previous research by Coleman et al (2010) and Ready et al (2013) that the NDRT may be closely associated with IQ. Relevant references are now included on lines 1008 – 1013.

Ready RE, Chaudhry MF, Schatz KC, Strazzullo S. "Passageless" administration of the Nelson-Denny reading comprehension test: Associations with IQ and reading skills. J Learn Disabil. 2013;46(4):377-384. doi:10.1177/0022219412468160

Coleman C, Lindstrom J, Nelson J, Lindstrom W, Gregg KN. Passageless comprehension on the Nelson-Denny reading test: well above chance for university students. J Learn Disabil. 2010;43(3):244-9. doi:10.1177/0022219409345017.

---

## [Editor Report · Decision Letter 2]

23 Jan 2024

Individual differences and the transposed letter effect during reading

PONE-D-23-23508R2

Dear Dr. Charlotte E Lee,

We’re pleased to inform you that your manuscript has been judged scientifically suitable for publication and will be formally accepted for publication once it meets all outstanding technical requirements.

Kind regards,

Montserrat Comesaña

Academic Editor

PLOS ONE

---

## [Editor Report · Acceptance letter]

19 Feb 2024

PONE-D-23-23508R2 

PLOS ONE

Dear Dr. Lee, 

I'm pleased to inform you that your manuscript has been deemed suitable for publication in PLOS ONE. Congratulations! Your manuscript is now being handed over to our production team.

Kind regards, 

on behalf of

Dr. Montserrat Comesaña Vila 

Academic Editor

PLOS ONE